# Disabilities in Early Childhood: A Global Health Perspective

**DOI:** 10.3390/children10010155

**Published:** 2023-01-12

**Authors:** Bolajoko O. Olusanya, Claudine Storbeck, Vivian G. Cheung, Mijna Hadders-Algra

**Affiliations:** 1Centre for Healthy Start Initiative, 286A Corporation Drive, Dolphin Estate, Ikoyi, Lagos 101223, Nigeria; 2Centre for Deaf Studies, University of the Witwatersrand, Johannesburg 999136, South Africa; 3Department of Pediatrics and Life Sciences Institute, University of Michigan, Ann Arbor, MI 48109, USA; 4University of Groningen, University Medical Center Groningen, Department of Pediatrics, Division of Developmental Neurology and University of Groningen, Faculty of Theology and Religious Studies, 9712 CP Groningen, The Netherlands

**Keywords:** developmental disability, developmental delay, global health, early intervention, early childhood development, ECDI2030, sustainable development goals

## Abstract

Prior to the launch of the United Nations’ Sustainable Development Goals (SDGs) in 2015, childhood disability was rarely considered an important subject in global health. The SDGs till 2030 now require that children under 5 years who are at risk of not benefitting from inclusive quality education are identified, monitored, and promptly supported. A new tool for identifying children who are not developmentally on track has been developed by UNICEF but has limited sensitivity for detecting children with disabilities due to reliance on parental assessment of child behavior in certain everyday situations. In this paper, we identified conditions that are commonly associated with developmental disabilities based on the International Classification of Diseases (ICD) codes and clarified the concept of “developmentally on track” as it relates to children with developmental disabilities and developmental delays. We summarized the latest evidence on the global burden of developmental disabilities in children under 5 years based on the diagnostic and functional approaches for measuring disabilities at the population level. We highlighted the global health context for addressing the needs of children with developmental disabilities and provided an overview of the opportunities and the role of pediatric caregivers in supporting children with developmental disabilities.

## 1. Introduction

The importance of early childhood development for the human capital development of a nation over the life course is widely acknowledged and embraced by professionals in various clinical and academic disciplines such as pediatrics, neurology, psychiatry, psychology, physiotherapy, speech and language pathology, rehabilitation medicine, public health, anthropology, sociology, education, and economics, and is now actively promoted as a global health agenda [1,2,3]. This multi-disciplinary and global recognition has been spurred lately by the United Nations’ Sustainable Development Goals, 2015–2030 (SDGs), which were launched in September 2015 [4]. One of the 17 SDGs (SDG 4) aims to ensure that all children have access to quality early childhood development, care, and pre-primary education so that they are ready for primary education by 2030 [4]. The SDG uniquely calls for the systematic monitoring of the proportion of children under 5 years who are developmentally on track in health, learning, and psychosocial well-being in every country. This approach allows for the identification of children who require special support to enable them to gain unrestricted access to inclusive quality education.

New tools, such as the Early Childhood Development Index 2030 (ECDI2030) by the United Nations Children’s Fund (UNICEF), have been developed to identify children who are “developmentally on track” primarily for statistical purposes [5]. The ECDI2030 captures the achievement of 20 key developmental milestones by children between the ages of 24 and 59 months based on mothers’ or primary caregivers’ opinions on how their children behave in certain everyday situations and the skills and knowledge they have acquired. However, the effectiveness of such tools for identifying all children with lifelong developmental disabilities is unclear. In this perspective paper, we shall explore why children with developmental disabilities are conceptually unlikely to be developmentally on track in the domains of health, learning, and psychosocial well-being compared to typically developing children. We shall also examine the global health framework for promoting early childhood development for children with developmental disabilities and the role of pediatric caregivers in ensuring that these children are promptly and adequately supported to optimize school readiness for inclusive education as envisioned by the SDGs.

## 2. Defining Developmental Disabilities

A major challenge confronting practitioners and policy makers in global health is the lack of consistency in the definition of developmental disabilities in the literature [6]. For example, it is not uncommon to find terms such as “mental, behavioral or neurodevelopmental disorders” [7], “developmental delays” [8], or “developmental differences” [9] used interchangeably with developmental disabilities. Perhaps, the most comprehensive definition is provided in -Section 102(8)(A) of the Developmental Disabilities Assistance and Bill of Rights Act of 2000, USA [10]. In this legislation, a developmental disability is defined as “a severe, chronic disability of an individual that: (i) is attributable to mental or physical impairment or a combination of mental and physical impairments; (ii) is manifest before the individual attains age 22; (iii) is likely to continue indefinitely; (iv) results in substantial functional limitations in 3 or more of the following areas of life activity (self-care, receptive and expressive language, learning, mobility, self-direction, capacity for independent living, and economic self-sufficiency); and (v) reflects the individual’s need for a combination and sequence of special, interdisciplinary, or generic services, individualized supports, and other forms of assistance that are of lifelong or extended duration and are individually planned and coordinated” [10]. An infant or a young child with developmental disability is further defined as “an individual from birth to age 9, inclusive, who has a substantial developmental delay or specific congenital or acquired condition, may be considered to have a developmental disability without meeting 3 or more of the criteria described in clauses (i) through (v) of subparagraph (A) above if the individual, without services and supports, has a high probability of meeting those criteria later in life” [10].

A more succinct definition adopted by the Center for Disease Control (CDC), USA describes developmental disabilities as a group of conditions due to an impairment in physical, learning, language, or behavior areas that begin during the developmental period (conception and birth to age 18 years) and usually impact day-to-day functioning throughout a person’s lifetime [11]. The conditions typically include cerebral palsy, epilepsy, hearing impairments (including deafness), speech or language impairments, visual impairments (including blindness), intellectual disability, autism spectrum disorder, and attention-deficit/hyperactivity disorder as shown in Table 1 [7,12,13].

The Individuals with Disabilities Education Act (IDEA), which is used as a mechanism for allocating federal funding for children with developmental disabilities in the USA, defines a child with a disability as “a child (i) with intellectual disabilities, hearing impairments (including deafness), speech or language impairments, visual impairments (including blindness), serious emotional disturbance, orthopedic impairments, autism, traumatic brain injury, other health impairments, or specific learning disabilities; and (ii) who, by reason thereof, needs special education and related services” [14].

These definitions are consistent with the United Nations’ Convention on the Rights of Persons with Disabilities which defines persons with disabilities to include “those who have long-term physical, mental, intellectual or sensory impairments which in interaction with various barriers may hinder their full and effective participation in society on an equal basis with others” [15]. According to the ICD codes, neurodevelopmental disorders are syndromes characterized by clinically significant disturbance in an individual’s cognition, emotional regulation, or behavior that reflects a dysfunction in the psychological, biological, or developmental processes that underlie mental and behavioral functioning [7]. These disturbances are usually associated with distress or impairment in personal, family, social, educational, occupational, or other important areas of functioning. The exclusion of sensory disorders from this category in the ICD suggests that neurodevelopmental disorders should be regarded as a subset of developmental disabilities (Table 1). It is also not uncommon to distinguish intellectual disability from other developmental disabilities under the term “intellectual and developmental disabilities” (IDD) to emphasize that developmental disabilities are not necessarily associated with intellectual disability [6]. While a developmental delay may be a red herring for a developmental disability, the two terms are not strictly interchangeable. A developmental delay assesses functioning in relation to general developmental milestones in typically developing children and can be constitutional, transitory, and self-limiting [16,17]. Unlike developmental delays, developmental disabilities are specific diagnostic entities that require a lifetime of support [18]. However, a developmental delay must be acted upon promptly. It is an unspecific sign that the complex process of development based on the interaction between the child, the child’s brain, and the environment is hindered. Developmental delay acts as an invitation to further diagnostics, such as fever which may signal different health conditions, including sepsis. Differentiating diagnostics in infancy may include a neurological exam [19]. This also means that when a developmental disability is clinically established, it must not be presented to parents as a developmental delay.

It is necessary to clarify that the ICD codes for the term “mental and behavioral dis-orders” are broad and can be associated with drug abuse or the (mis)use of narcotics. The term must, therefore, be used cautiously for children under 5 years.

## 3. Measuring and Counting Developmental Disabilities

For governments and health authorities to introduce and implement policies to support children with developmental disabilities, it is necessary to determine how many children are affected by these conditions in any population at any given period. For many years, children with developmental disabilities have been overlooked in global health because of the absence of population-based data to inform policy decisions. A major challenge has been associated with the difficulties in measuring disabilities and generating comparable data across different disabilities and geographical locations globally [20,21,22].

Measuring disability is closely linked with the conceptualization of disability and has resulted in two main approaches. The first one is the diagnostic approach which relies on the medical model of disability, which views disability as a medical phenomenon that results from impairments in body functions or structures: a deficiency or abnormality [23]. This measurement approach involves identifying the number of children with specific disabilities in a clinical or educational setting and is typically based on the ICD codes from the World Health Organization (WHO) [7] or the American Psychiatric Association’s Diagnostic and Statistical Manual of Mental Disorders (DSM) [24]. The use of diagnoses and labeling to determine which individuals receive services (e.g., educational services), types of services, and benefits have been criticized for placing enormous power and authority in the medical personnel over the lives of the social perception of individuals with disabilities in society [23]. Moreover, little consideration is given to the preferences of the affected individuals, including children with disabilities and their families [23].

The second approach is associated with the bio-psychosocial model of disability, which combines the medical and social models of disability [25]. The biopsychosocial model was developed to address the limitations of the medical model in recognizing the psychological, social, and behavioral dimensions of a medical condition and has been adopted as the framework for the WHO’s International Classification of Functioning, Disability, and Health (ICF) [26]. The ICF conceptualizes a person’s level of functioning as a dynamic interaction between her or his health conditions, environmental factors, and personal factors and provides a standard language for the definition and measurement of health and disability. The measurement approach based on ICF typically involves the use of questionnaires administered to mothers and primary caregivers in household surveys to identify children with functional difficulties in the developmental domains of hearing, vision, mobility, fine motor, communication/comprehension, emotions, learning, and playing. A major limitation of this approach is the poor sensitivity in identifying children with mild degrees of functional impairments who would require support to optimize their development and well-being as reflected in the ICF.

## 4. Prevalence and Burden of Developmental Disabilities

The global prevalence of children with developmental disabilities using both the diagnostic approach and functional approach has been reported [22,27,28]. In 2018, based on the diagnostic approach, the Global Burden of Disease (GBD) Study estimated that 53 million children under 5 years globally had mild-to-severe developmental disabilities in 2016, including epilepsy, intellectual disability, sensory impairments, autism spectrum disorder, and attention-deficit/hyperactivity disorder, and 95% resided in LMICs [27]. In 2021, using the functional approach, UNICEF estimated that 236 million children and adolescents (younger than 18 years) have moderate-to-severe disabilities globally, among whom were 29 million children aged 0–4 years [28].

A comparative analysis of the GBD and UNICEF databases suggested that while the categories of developmental disabilities differed significantly, the prevalence estimates for children and adolescents were statistically equivalent but not for children under 5 years [22]. The major reason for the difference among children under 5 years was the difficulty in eliciting functional difficulties among children under 2 years of age using parent-reported responses.

UNICEF report has further shown that children with disabilities are 42% less likely to have foundational reading and numeracy skills, 49% more likely to have never attended school, 47% more likely to drop out of primary school, 32% more likely to experience severe physical punishment at home, and 20% less likely to have expectations of a better life, compared to children without disabilities [28]. Children with disabilities are also at a higher risk of common causes of under-5 mortality, such as acute respiratory infection, fever, diarrhea, and malnutrition.

It is important to clarify that the reported data by UNICEF and GBD Study would appear as the best possible estimates based on the approaches adopted by the researchers. The reports are likely to have underestimated the true prevalence of developmental disabilities globally in view of the limitations associated with the methods. For example, the GBD estimates are based on six developmental disabilities, while the UNICEF estimates excluded children with a mild but functionally significant degree of disability. Notwithstanding, the available evidence clearly shows that developmental disabilities are highly prevalent among children in early childhood and are associated with poor health and educational outcomes. The affected children, therefore, deserve to be prioritized in any global health agenda or initiative for early childhood development. In effect, the needs of children with developmental disabilities should in practical terms receive greater and special attention from policy makers and caregivers than the needs of children without disabilities.

## 5. Clarifying the Concept of Early Childhood

Investing in the early years helps to break the cycle of poverty, address inequality, and boost productivity. The period considered “early childhood” typically varies from the first 1000 days beginning at conception or birth to the period from birth up to age 8 years [29]. Since developmental disabilities may manifest between birth and age 22 years, it is necessary to determine the period during which intervention is most critical to optimize support for the affected individuals. Extensive evidence from neuroscience has shown that the first five years of life are the fastest period of human growth and development, as 90 percent of a person’s brain development occurs by the age of five [30,31,32]. The most dramatic developmental changes occur prenatally and during the first two years after term age (Figure 1). This means that the high neuroplasticity present during the first two postnatal years offers the best opportunity to optimize functional outcomes in children with developmental disabilities, with further opportunities offered in the following three years. The misunderstanding of this scientific underpinning of human brain development frequently manifests in the poor conceptualization of early intervention for optimal early childhood development in global health.

In the SDGs, the flagship global health agenda, early childhood is defined to include all children younger than 5 years. This threshold reflects school entry at age 5–6 years and the beginning of early childhood care in most educational settings. It is not uncommon to find national intervention programs designed separately for children aged 0–3 years and the preschool period, 4–5 years [14,33]. Some researchers have also drawn attention to the relationship between the first 1000 days (typically, 0–2 years) used in many early childhood development programs [29] and the next 1000 days (2–5 years) [34]. While intervention in the first 1000 days is often associated with the best developmental outcomes, the next 1000 days are considered a critical period for reinforcing the gains in the first 1000 days and establishing healthy developmental trajectories in young children [34]. More crucially, the first five years of life provide the opportunity to monitor the trend in developmental disabilities at the population level globally [35] and to compare it with the trends in other global health indicators in the SDGs, such as stunting, malnutrition, and under-5 mortality [4,36]. For example, under-5 mortality conceptually reflects the social, economic, health, and environmental conditions in which children live [27,35,36].

## 6. The Importance of Global Health

The history and concept of global health are multi-dimensional, and a full discussion is beyond the scope of this paper [37]. The predominant concept involves the mobilization of technical and financial resources in the Global North to address health problems in the Global South, where 90% of the world’s children reside. Within this framework, the key players in the Global North include governments, philanthropists, charitable organizations, and public-health experts who often agree on a common agenda to guide the deployment of resources in the form of developmental assistance for health (DAH) to beneficiaries in the Global South. This funding support is often channeled through bilateral organizations such as UNICEF and WHO or directly to governments and international civil society organizations. Very rarely does the Global North pay attention to or prioritize issues outside of the global health agenda except in the case of a pandemic or a humanitarian emergency resulting from a natural disaster or armed conflict. It is also unusual for the Global North to promote or implement childcare and development programs designed for the Global South for their population [29].

The health systems in many countries in the Global South are highly dependent on DAH to deliver essential services to their citizens. For example, there is a global interest in bridging the health inequalities between the Global North and the Global South, which is encapsulated in the SDG provision for universal health coverage (UHC) [4]. UHC provides the opportunity for children with developmental disabilities to access a full spectrum of essential, quality health services, from health promotion to prevention, treatment, rehabilitation, and palliative care across the life course [38]. The unprecedented focus on early childhood development towards inclusive education in the SDGs has also provided a pathway for addressing the needs of children with developmental disabilities as a global health imperative. This is reinforced by specific disability-inclusive provisions in five out of the 17 SDGs, in sharp contrast with the Millennium Development Goals which excluded such provisions [4].

The role of pediatricians in global health has been elucidated in a policy statement issued by the American Academy of Pediatrics in 2018 [39]. The statement makes explicit reference to the SDGs as the link to global health for pediatricians. Not all pediatricians are involved with or knowledgeable about global health, which has now emerged as a full-fledged specialty with a training curriculum. Those working in global health are enjoined to work collaboratively with international partners to achieve child-related SDGs and the United Nations’ Convention on the Rights of the Child. This would entail advocating for policies and investments that support global child health, including the optimal development of every child and equitable access to quality education in line with the prevailing global health agenda and commitments [39].

The ability to frame health issues that are supported by high-quality evidence in a manner that can be readily appreciated and acted upon by decision or policy makers is also essential [3,40]. It is also important to distinguish “global health” from “international health” [37]. While global health can be considered an extension of international health, the latter does not necessarily involve participation from the Global South. In global health, participation based on equity, equality, diversity, and inclusion is essential for facilitating a sense of ownership and commitment to global programs in the target population [3].

## 7. Supporting Children with Developmental Disabilities

The benefits of early intervention in maternal and child care are substantially based on empirical and scientific evidence [30,31,41]. Early detection and intervention for children with developmental disabilities is already a public health imperative in high-income countries [13,14,33,42,43]. These services have been proven to optimize school readiness for children with developmental disabilities and are a prerequisite for achieving inclusive education as envisioned by the SDGs [44,45]. Poor or a lack of access to education places children, especially girls with developmental disabilities, at a greater disadvantage in securing gainful employment [46]. The eradication of extreme poverty (SDG 1.1); ending all forms of malnutrition, including the internationally agreed targets on stunting and wasting in children under 5 years of age (SDG 2.2); and ending preventable deaths of newborns and children under 5 years of age (SDG 3.2) by the year 2030, are major priorities for all countries, particularly in the Global South. Children with developmental disabilities have been shown to be the most vulnerable population in these core areas of global health concerns but are still not prioritized for appropriate support [38,47,48,49]. For example, despite the strong association between disability and malnutrition, children living with disabilities are often marginalized in malnutrition research and treatment programs, as well as current international guidelines for severe acute malnutrition [48]. Similarly, disability is rarely recognized as a risk factor for child mortality in global health [36].

Several barriers to addressing the needs of children with disabilities have been reported, which include stigma and discrimination, lack of a trained workforce, and ill-equipped health systems [50,51,52,53]. WHO has identified 40 targeted actions for addressing these barriers and achieving equity for persons with disabilities grouped under 10 key areas: political commitment, leadership, and governance; health financing; engagement of stakeholders and private sector providers; models of care; health and care workforce; physical infrastructure; digital technologies for health; systems for improving the quality of care; monitoring and evaluation; and health policy and systems research [54]. Pediatric caregivers, including nurses, physicians, other primary care professionals, community health workers, and rehabilitation specialists, have a significant role in promoting school readiness for children with or at risk of developmental disabilities through pediatric consultations as well as advocacy [55,56]. They must work in partnership with parents and other family members where necessary to ensure that the services being provided are appropriate and acceptable. Recognizing the challenge of a limited workforce for delivering such services to children with disabilities in early childhood, the International Pediatric Association has issued a position statement that emphasizes the urgent need and pathways to the training of pediatricians and other ancillary care providers [56]. WHO has also developed a caregiver skills training toolkit for families of children with developmental delays or disabilities, which is currently being piloted in some countries [57].

As previously reported, global health organizations such as WHO, UNICEF, and the World Bank have a critical role to play in supporting countries to promote early childhood interventions for children with disabilities and facilitate the realization of the ambitious goal for inclusive and equitable quality education for all by 2030 [4,58]. The available guidelines for early childhood development from these organizations are quite limited in addressing the complex and diverse needs of children with disabilities [29,59]. It is not unlikely that these gaps will be addressed comprehensively in the forthcoming WHO-UNICEF Global Report on Children and Young People with Developmental Disabilities [60].

## 8. Conclusions

Developmental disabilities are highly prevalent globally, and the affected children are constitutionally unlikely to be developmentally on track in health, learning, and psychosocial well-being compared to children without disabilities. The disability-inclusive provisions in the SDGs, including the emphasis on early childhood development for inclusive education, have provided a pathway for children with developmental disabilities to be accorded priority in the global health agenda till 2030. Priority consideration goes beyond symbolic reference to children with disabilities in policy documents simply to portray inclusiveness. It must be reflected in the share of resource allocation for early childhood development globally. Pediatric caregivers, in partnership with parents and other family members, have a critical role in ensuring school readiness for inclusive education among children with developmental disabilities.

## Figures and Tables

**Figure 1 children-10-00155-f001:**
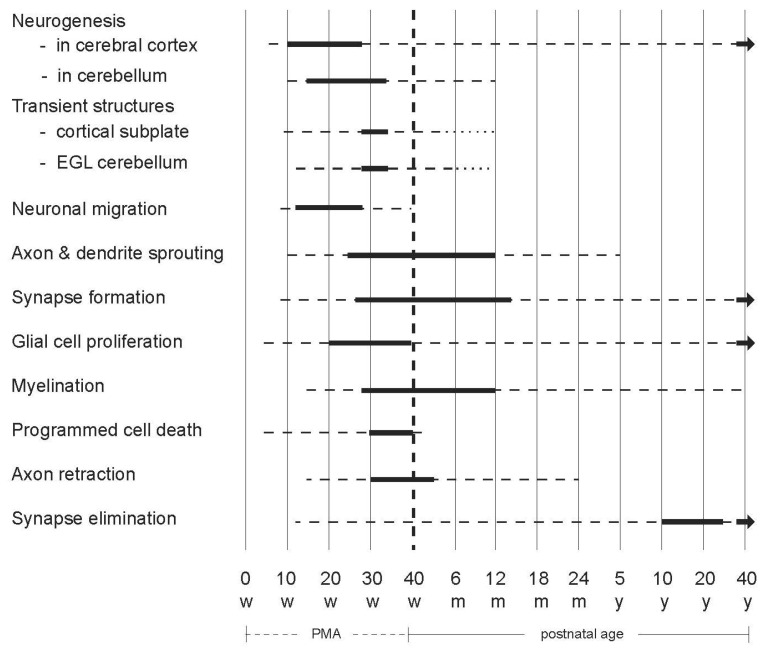
Schematic overview of the developmental processes occurring in the human brain (based on [31]). The bold lines indicate that the processes mentioned on the left side are very active, the broken lines denote that the processes continue but less abundantly. EGL = external granular layer; m = months; PMA = postmenstrual age; w = weeks; y = years. Figure reproduced with permission from ‘Early Detection and Early Intervention in Developmental Motor Disorders—from neuroscience to participation’ by Mijna Hadders-Algra (ed.) published by Mac Keith Press in its Clinics in Developmental Medicine Series, ISBN number 978-1-911612-43-8.

**Table 1 children-10-00155-t001:** Major Categories of Developmental Disability with Corresponding ICD-11 Diagnostic Codes.

CONDITIONS	ICD CODES
**Cognitive**	
Intellectual Disability	6A00
Mild (IQ approximately 50–69)	6A00.0
Moderate (IQ approximately 35–49)	6A00.1
Severe (IQ approximately 20–34)	6A00.2
Profound (IQ below 20) F73	6A00.3
Developmental Learning Disorder	6A03
With Impairment in Reading (Dyslexia)	6A03.0
With Impairment in Written Expression	6A03.1
With Impairment in Mathematics (Dyscalculia)	6A03.2
With Other Specified Impairment of Learning	6A03.3
**Motor**	
Developmental Motor Coordination Disorder	6A04
Cerebral Palsy	8D20, 8D21, 8D22, 8D2Y, 8D2Z
Post-Polio Progressive Muscular Atrophy	8B62
Muscular Dystrophies	8C70
Spina Bifida	LA02
Spinal Muscular Atrophies	8B61
**Neurological**	
Epilepsy or Seizures	8A60.0Y, 8A60.1–8A60.9
**Vision**	
No Vision Impairment	9D90.0
Mild Vision Impairment	9D90.1
Moderate Vision Impairment	9D90.2
Severe Vision Impairment	9D90.3
Blindness	9D90.4, 9D90.5
**Hearing**	
Congenital Hearing Impairment	AB50, AB50.0, AB50.1, AB50.2
Acquired Hearing Impairment	AB51, AB51.0, AB51.1, AB51.2
Deafness	AB52
**Developmental Speech or Language Disorders**	
Developmental Speech Sound Disorder	6A01, 6A01.0, 6A01.1
Developmental Language Disorder	6A01.2, 6A01.20–6A01.23
**Behavior**	
Attention-Deficit Hyperactivity Disorder	6A05, 6A05.0–6A05.2
Autism Spectrum Disorder	6A02, 6A02.0–6A02.5

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
