# Peer review of "Disabilities in Early Childhood: A Global Health Perspective"

_children, 2023, doi:10.3390/children10010155_

Round 1

Reviewer 1 Report

Introduction and definitions are well done -- establishes the importance of the topic and comprehensively looks at various international definitions, which is necessary given the vagueness with which it is often defined in the literature. You did a good job in arguing the importance of not only catching developmental disabilities early but also acting on them. You addressed common barriers and argued the need for improving the pediatric caregiver workforce, who plays a critical role in helping advance the WHO goals.

Although you explained that developmental delays are not the same as developmental disabilities, there is a great challenge in differentiating whether a particular child simply has delays in development or whether their lower achievement (for example, in speech-language development) constitutes a disorder, particularly in earlier childhood. With already limited resources, how do LMICs reconcile how to allocate personnel and financial resources only to those with disabilities and not those with delays? Is that even possible to do? If it is possible, is it something they should do? Or is it equally important to identify and support those with delays as it is for those with permanent disabilities?

I appreciate your efforts to give estimates of the burden of developmental disabilities globally. Any thoughts as to whether the numbers may be underestimates, given the poor access to diagnostic services (for example, to diagnose autism spectrum disorder) in many LMI countries?

line 159-165 needs a citation. It refers to UNICEF but does not indicate from which UNICEF document it came from.

Author Response

Although you explained that developmental delays are not the same as developmental disabilities, there is a great challenge in differentiating whether a particular child simply has delays in development or whether their lower achievement (for example, in speech-language development) constitutes a disorder, particularly in earlier childhood. With already limited resources, how do LMICs reconcile how to allocate personnel and financial resources only to those with disabilities and not those with delays? Is that even possible to do? If it is possible, is it something they should do? Or is it equally important to identify and support those with delays as it is for those with permanent disabilities?

Reply: We thank the reviewer for this important comment. We have provided additional clarification in the text to address this point [see lines 126-132].

I appreciate your efforts to give estimates of the burden of developmental disabilities globally. Any thoughts as to whether the numbers may be underestimates, given the poor access to diagnostic services (for example, to diagnose autism spectrum disorder) in many LMI countries?

Reply: We agree that the reported data either using a functional approach or the medical approach are underestimates. As the reviewer rightly observed both approaches are associated with limitations. This clarification has now been added to the text [see lines 195-200].

line 159-165 needs a citation. It refers to UNICEF but does not indicate from which UNICEF document it came from.

Reply: We thank the reviewer for this observation. The reference [#28] has now been added [lines 188-192].

Reviewer 2 Report

The authors present a manuscript describing global health efforts to identify and serve preschool children with developmental disabilities. The information and insights regarding definitions of developmentally delayed and developmental disabilities is helpful. A global estimate is provided. 

One question about the conclusion- yes, it is true that "pediatric caregivers" have a critical role in ensuring school readiness for inclusive education. But how do you see parent caregivers involvement? They would need education on how to proceed with this action.  How do you see this as part of the process?

Author Response

One question about the conclusion- yes, it is true that "pediatric caregivers" have a critical role in ensuring school readiness for inclusive education. But how do you see parent caregivers involvement? They would need education on how to proceed with this action.  How do you see this as part of the process?

Reply: We thank the reviewer for drawing attention to the role of parent caregivers. We have now provided further clarification in the text [lines 314-321, 343-344].

Reviewer 3 Report

The article presents a narrative review, and addresses an interesting aspect, such as early disability. I do not consider the methodology used to be published, since I understand that a review must be systematic, or meta-analysis, with its Pedro, Prisma, or other criteria. These criteria endorse the quality and rigor of the chosen articles, and the bibliographic search. This opinion does not believe that the article is of poor quality, but it does not follow a scientific methodology.

As a theoretical approach to the subject it is complete, but it cannot be considered a meta-analysis article on the subject, nor have the sections on results, discussion, limitations, etc. necessary for a scientific article been taken into account.

Author Response

The article presents a narrative review, and addresses an interesting aspect, such as early disability. I do not consider the methodology used to be published, since I understand that a review must be systematic, or meta-analysis, with its Pedro, Prisma, or other criteria. These criteria endorse the quality and rigor of the chosen articles, and the bibliographic search. This opinion does not believe that the article is of poor quality, but it does not follow a scientific methodology. As a theoretical approach to the subject it is complete, but it cannot be considered a meta-analysis article on the subject, nor have the sections on results, discussion, limitations, etc. necessary for a scientific article been taken into account.

Reply: We thank the reviewer for this observation. Our study is neither a systematic review nor a meta-analysis. Like most journals, Children (Basel) makes provision for non-systematic reviews which are essentially narrative reviews usually written by subject experts. In some journals, our review may be classified as a discussion paper, perspective, expanded editorial, or commentary. However, these nomenclatures are not available in this journal which left us with the current option of a “non-systematic/narrative review”. While there are prescribed sections for narrative reviews besides the introduction, the key messages in each section have been supported with carefully selected and up-to-date references as evidence.

Round 2

Reviewer 3 Report

I have no further comment.